Effect of cigarette smoking on the optical properties of contemporary dental ceramics: an in-vitro analysis

Alqahtani Abdulaziz 1
AlHelal Abdulaziz A. 1
Albani Ragad 1
Ali Mohsin 2
Badghshar Omar Ali Omar 3
Khan Aleshba Saba 4
Habib Syed Rashid syhabib@ksu.edu.sa rashidhabib@hotmail.com 1
1 Department of Prosthetic Dental Sciences, College of Dentistry, King Saud University , Riyadh , Saudi Arabia
2 Department of Comprehensive Dentistry, School of Dental Medicine, Case Western Reserve University , Cleveland , OH , United States of America
3 College of Dentistry, King Saud University , Riyadh , Saudi Arabia
4 Department of Prosthodontics, Shahida Islam Dental College , Lodhran , Pakistan
Uversky Vladimir
Electronic publication date: 2024 Nov 27
Publication date: 2024
Volume: 12
Electronic Location ID: e18564
Received 2024 Aug 23; Accepted 2024 Oct 31
Copyright: ©2024 Alqahtani et al.
Copyright year: 2024
Copyright holder: Alqahtani et al.
License: This is an open access article distributed under the terms of the Creative Commons Attribution License, which permits unrestricted use, distribution, reproduction and adaptation in any medium and for any purpose provided that it is properly attributed. For attribution, the original author(s), title, publication source (PeerJ) and either DOI or URL of the article must be cited.
License URL: https://creativecommons.org/licenses/by/4.0/

Keywords: Cigarettes smoking, Dental shade, Dental Ceramics, Spectrophotometer, Tooth shade

Funding: Researchers Supporting Project RSPD2024R950 King Saud University, Riyadh, Saudi Arabia This research was funded by the Researchers Supporting Project (RSPD2024R950), King Saud University, Riyadh, Saudi Arabia. The funders had no role in study design, data collection and analysis, decision to publish, or preparation of the manuscript.

==============================
Background

Cigarette smoking is the most common form of tobacco use worldwide. With the frequent introduction of new dental materials, the effect of smoking on their optical properties such as long term color stability, should to be thoroughly investigated.

Objective

This in-vitro study aims to investigate the effect of smoking on the optical properties of contemporary dental ceramics used currently for restoration of teeth.

Methods

Five different materials in two shades (B1 and C1) were used with 15 samples from each pressable lithium disilicate (Emax), layered lithium disilicate (Lmax), porcelain fused to metal (PFM), monolithic zirconia (MZr) and layered zirconia (LZr) were used (n = 75). The samples were exposed to conventional cigarette smoke and color stability was assessed at four different time intervals i.e., baseline, 1 week, 1 month and 6 months. CIELAB color space (CIE L*a*b*) values were used to evaluate the color difference (ΔE). A one-way analysis of variance (Anova) was used for statistical analysis of ΔE. Significant P-value was kept as <0.05, followed by Tukey post-hoc test.

Results

All test materials demonstrated significant color differences (ΔE) after exposure to cigarette smoke (p < 0.05). For shade B1, the highest change in shade ΔE 17.02 was exhibited by Lmax, whereas the least change in shade was exhibited by Emax followed by PFM at values of ΔE 10.11 and 11.2 respectively. For shade C1, the highest change (11.47) in shade at 6 months was demonstrated by MZr, whereas lowest values of ΔE were exhibited by Emax (7.52).

Conclusions

Traditional smoking causes significant change in shade of dental ceramics which can affect the esthetics of the patients. All material samples tested showed the values of ΔE > 3.3 which is higher than the acceptable range. Lowest color change was observed in Emax and PFM.

Introduction

There has been an increased demand for esthetic dental procedures due to enhanced awareness of general public for facial esthetics (Alrabeah et al., 2023a). Recent advancements in dental materials (Alrabeah et al., 2023a; Blatz et al., 2019) are also directed towards achieving this goal. Dental ceramics have been the material of choice for tooth colored indirect restorations for both posterior and anterior teeth (Alrabeah et al., 2023a; De Matos et al., 2022). These restorations are well known for their exquisite appearance and promising mechanical strength (De Matos et al., 2022; Warreth & Elkareimi, 2020). However, the harsh oral environment is a continuous challenge for teeth and restorations to maintain the desired esthetics (Alrabeah et al., 2023a; Lamont, Koo & Hajishengallis, 2018). There is a continuous exposure of hot and cold beverages and stain causing stimuli in the mouth (Lamont, Koo & Hajishengallis, 2018).

Undoubtedly the cigarette smoking is injurious to health in general and specifically to oral cavity (Alrabeah et al., 2023a; Tanthanuch et al., 2022; World Health Organization, 2023). It can also effect esthetics of teeth and restorations by causing staining of enamel, dentin and restorations (Alrabeah et al., 2023a; Karanjkar et al., 2023). A study by Dalrymple et al. (2021) reported that smoking can result in staining of dental ceramics. The smoke emerging from a burning cigarette is considered as the whole smoke and is comprised of small droplets suspended in a combination of semi-volatile compound and gases (Karanjkar et al., 2023). When this is without nicotine, it is termed as tar which tends to collect on end of cigarette filter and causes it to turn yellowish brown (Karanjkar et al., 2023; Zanetti et al., 2019). Thus tar is thought to be the culprit of staining the teeth and restorations (Zanetti et al., 2019). Apart from tar, there are eleven other compounds identified from tobacco that lead to staining of teeth and restorations (Karanjkar et al., 2023). Although limited evidence is available to determine the exact mechanism causing discoloration in teeth and ceramic restorations, an analysis of the microscopic findings show structural destruction of enamel including rods, pits and holes at variable levels (Ibrahim & Hassan, 2021). This destruction leads to irregular surface which can cause staining (Ibrahim & Hassan, 2021). A study by Al Moaleem et al. (2022), confirmed that tobacco causes surface changes in a restoration that leads to staining.

Most widely used indirect restorative materials currently in dentistry are ceramics including the pressable and layered Emax ceramics, porcelain fused to metal, monolithic and layered zirconia (Schelkopf et al., 2022). The reason for their increased acceptance is the amalgamation of esthetics, biocompatibility and favorable mechanical properties (Schelkopf et al., 2022). Color stability of materials requires a smooth surface at micro level as rough surface tends to collect more stains (Schelkopf et al., 2022). Translucency and opacity of dental ceramics can be affected by changes in surface roughness thus altering the color of the restoration (Al Moaleem et al., 2022). The glaze on surface of ceramics reduces the adherence of extrinsic stains and helps in stability of the shade (Al Moaleem et al., 2022). A study investigated the effect of tobacco on different restorations in oral cavity concluded that smoke causes abrasive effect on restorations when measured objectively (Thompson et al., 2016). Tobacco leads to roughening of surface of the restoration at micro level by abrasive affect, thus, the restorations with less hardness tend to take up more stains (Thompson et al., 2016; Makkeyah et al., 2024).

Although dental ceramics appeared to have a bright future, the increased use of cigarette smoking globally necessitates more research into the effect of tobacco on their color stability and optical properties. This in vitro study is a spectrophotometric analysis of the effect of cigarette smoking on the optical properties of ceramics using CIELAB coordinates. The aim is to help clinicians in the selection of esthetic materials that are resistant to staining for patients with the history of heavy cigarette smoking.

Materials and Methods

The research was carried out at King Saud University’s College of Dentistry’s Department of Prosthodontics and College of Dentistry Research Center (CDRC) in Riyadh, Saudi Arabia. Approval from ethical board of the university was obtained before starting the research work (Registration No PR-0128).

Materials used

Two shades (B1 and C1) of five commonly used dental ceramics were included in this study. These were pressable ceramics (Emax), layered ceramics (Lmax), porcelain fused to metal (PM), monolithic zirconia (ZrM) and layered zirconia (ZrL). Fifteen specimens of each material were fabricated as discs of 10 mm diameter and two mm thickness using appropriate conventional and CAD/CAM techniques. The entire sample size (n = 75) was calculated using the G*Power software (G*Power 3.1.9.7; G*Power, Dusseldorf, Germany) with an effect size of 0.45, a power of 0.85, and a significance threshold of 0.05 or less. Discs were cleaned, polished, and their color was measured using a spectrophotometer at baseline and after being subjected to cigarettes smoke for a period of 1 week, 1 month and 6 months in a custom designed smoking chamber. The details of all the test materials are presented in Table 1.

Specimen preparation

All the specimens from each group were prepared as per the instructions of manufacturer. An electronic caliper was used to standardize the thickness of disks to 10 mm diameter and two mm thickness (Fig. 1). Computer aided design/computer aided machine (CAD/CAM) was used for preparation of milled ceramic disks. 3D printer was used to prepare metal core of PFM specimens and ceramic veneer was added by an expert technician. All the specimens were prepared according to the manufacturers’ instructions and finally glazed with glaze material (IPS e.max Essence; Ivoclar Vivadent). Each disk was analyzed for acceptable quality by two specialists after preparation.

Table 1 Information on the test materials used in the study.

S. no.	Symbols	Ceramics	Trade information	Company details	
1.	Emax	Pressed	IPS e.max Press	Ivoclar Vivadent, AG Schaan, Liechtenstein.	
2.	Lmax	Pressed	IPS e.max Press		
Layering	IPS InLine Dentin		
3.	ZrL	Zirconia base	ZI, LT, Zolid	Amann-Girrbach, Koblach, Austria.	
Ceramic layer	IPS Inline Dentin	Ivoclar Vivadent, AG Schaan, Liechtenstein.	
4.	ZrM	Block	Zolid gen-x	Amann-Girrbach, Koblach, Austria.	
Shade of the frame	Dying liquid	3M ESPE, USA	
5.	PM	Coping metal	Starbond easy powder 30	Scheftner dental alloys, S&S GmbH, Germany	
Layering	IPS InLine Dentin	Ivoclar Vivadent, AG Schaan, Liechtenstein.	
Information about the Cigarettes used for exposing the specimens to smoking.	
6.	Smoke	Cigarettes	Parliament	Parliament silver blue, Philip Morris, Switzerland.	

Figure 1 Picture of some of the ceramic specimens used in the study.

Thermocycling of prepared disks

To replicate the clinical condition of oral environment, all the disks were kept in distilled water for 24 h. Water temperature was maintained at 37 °C. Aging of disks was done for 6,000 rotations between 5 °C and 55 °C in a thermocycler (SD mechatronic; Huber thermocycler, Berching, Germany). Dwelling time was 30 s and 5 s was kept as transfer time.

Conventional cigarette smoke exposure

Conventional cigarettes (Parliament silver blue; Philip Morris, Lausanne, Switzerland) with Nicotine of 0.3 mg concentration were used for the exposure of the ceramic specimens to its smoke. A vacuum system reproducing artificial lung was devised which directed the smoke in a custom made plastic chamber (Fig. 2). The chamber was a glass box with 10 × 10 × 10 dimensions. Two vents were cut in two parallel vertical walls of the chamber; one for attaching the external vacuum device and the other one for cigarettes. The disks of different materials were held by polyvinyl silicone putty (3M™ Express™ XT Putty Soft VPS) placed at distance of 10 cm from incoming smoke inside the chamber. The disk samples were kept in straight position relative to both sides of the compartment to replicate oral environment. Cigarette was fixed into chamber at one end. Negative pressure was created to direct the smoke coming out from the cigarette in to the chamber. Cigarettes were left to burn for 10 min. Specimens were subjected to smoke for a period of 1 week, 1 month and 6 months with average of 10 cigarettes/day.

Figure 2 Customized smoking chamber used for the smoke exposure (1, Ambu Bag; 2, the clear; chamber; 3, samples; 4, cigarette; 5, smoke).

Spectrophotometric analysis

Spectrophotometer (Labscan XE; Hunter Lab, Reston, VA, USA) was used for evaluation of color change ΔE in terms of L*a*b values (Commision Internationale de l’Eclairage). L* is for lightness, a* for axes of colors i.e., −a shows green, +a shows red, −b shows blue and +b shows yellow axes. Each specimen was kept in front of a black background and the sensor of spectrophotometer was placed above it. Three values were recorded for every specimen (Fig. 3).

The formula used to determine the color stability was: Color differenceΔE=ΔL∗2+Δa∗2+Δb∗21/2.

ΔL, Δa and Δb were calculated as difference in different time exposures and baseline readings as below;

ΔL = L (Different time exposures) − L (Baseline reading);

Δa = a (Different time exposures) − a (Baseline reading);

Δb = b (Different time exposures) − b (Baseline reading).

Color assessment

L*a*b* values were acquired for each specimen at baseline and immediately after the last smoking cycle of each interval. All measurements were repeated three times, the means of the L*a*b* values were measured and the color difference ΔE (Smoking-Baseline) was calculated using the equation given above, where ΔE is the color difference and ΔL*, Δa*, Δb* represented changes in lightness, red–green coordinate, and yellow-blue coordinate, respectively.

Figure 4 presents the flow chart of the methodology followed in the present study.

Figure 3 A screen shot of the recorded L*a*b* values in the spectrophotometer.

Figure 4 Flow chart of the methodology used in the study.

Statistical tests

Statistical Package for the Social Sciences (SPSS) version 23 was used to analyze the data. For each sample of sample disk, the descriptive variables measured the L*a*b* mean value and ΔE for these values. One-way ANOVA and Tukey’s post Hoc were used for comparison of mean for ΔE. Confidence interval was kept as 95% for each sample. Level of significance was kept as <0.05.

Results

The descriptive statistics (standard deviation and mean) for L* value for shade B1 and C1 of five materials (N = 75) are given in Table 1. For the shade B1 (Table 1) at baseline, the layered ceramics (Lmax) showed the lowest mean value (63.00) and PFM has the highest value (76.19). Pressable and layered ceramics as well as layered zirconia had closer values but monolithic zirconia and PFM showed greater difference even at the baseline readings. A common observation for all the materials was that the lightness value continuously reduced with exposure at 1 week, 1 month and 6 months.

For the shade C1 (Table 2) at baseline, pressable ceramics showed lowest mean value (51.45) and PFM showed the highest value (73.25). The lightness value increased for all materials at 1 week exposure compared to baseline value. At exposure for 1 month and 6 month, the value reduced significantly and even decreased from baseline value. The pattern was different for pressable ceramic only, which increased at 1 week then reduced at 1 month and increased again at 6 months. At 6 month exposure, the least value observed was for layered ceramic (54.04) and highest was for PFM (65.48).

Table 2 Standard deviation (S.D) and mean of L* values of five test materials (shade B1 & C1).

Shade	Time interval	Emax	Lmax	PFM	ZrM	ZrL	
		Mean	S.D	Mean	S.D	Mean	S.D	Mean	S.D	Mean	S.D	
B1	Baseline	64.07	0.54	63.00	1.25	76.19	0.54	72.63	0.50	65.88	1.15	
1 week	61.49	0.63	61.66	0.80	72.99	0.22	69.88	0.66	64.54	0.88	
1 month	56.84	0.52	60.40	1.10	72.06	0.63	69.66	0.51	62.77	0.99	
6 months	55.11	0.58	47.43	3.30	65.97	0.54	61.66	1.55	56.81	1.46	
Mean	59.38	3.65	58.12	6.56	71.80	3.76	68.46	4.23	62.50	3.67	
C1	Baseline	51.45	0.70	59.24	1.43	73.25	0.50	69.10	0.97	63.34	0.84	
1 week	54.58	0.29	63.46	1.51	75.07	0.53	71.74	2.25	67.80	2.99	
1 month	52.19	0.34	58.61	0.84	69.85	0.32	65.67	3.15	62.10	0.77	
6 months	64.88	0.43	54.04	1.55	65.48	0.65	60.11	0.70	59.61	3.89	
Mean	55.77	5.45	58.84	3.62	70.91	3.72	66.65	4.81	63.21	3.87	

The descriptive statistics (standard deviation and mean) for a* value for shade B1 and C1 of five materials (N = 75) are given in Table 2. For the shade B1 (Table 3) the observations are in green zone at baseline. The general trend shows increase in values thus change from green towards red zone. The largest increase in values were seen after 6 months in all samples of shade B1. Out of five materials, the largest increase in value was seen for layered ceramics after 6 months (mean = 5.50).

Table 3 Standard deviation and mean of a* values of five test materials (shade B1 & C1).

Shade	Time interval	Emax	Lmax	PFM	ZrM	ZrL	
		Mean	S.D	Mean	S.D	Mean	S.D	Mean	S.D	Mean	S.D	
B1	Baseline	−1.07	0.05	−0.73	0.14	−0.52	0.25	−0.82	0.27	−0.05	0.18	
1 week	−0.68	0.09	−0.39	0.73	0.26	0.39	−0.21	0.30	0.28	0.77	
1 month	−0.11	0.04	−0.47	0.75	0.59	0.58	−0.13	0.29	0.38	0.21	
6 months	2.64	0.49	5.50	1.54	3.06	0.85	3.42	0.58	2.77	0.60	
Mean	0.19	1.48	0.98	2.75	0.85	1.46	0.57	1.73	0.84	1.18	
C1	Baseline	−1.186	0.03	−1.27	0.10	0.64	0.40	−1.22	0.36	−0.76	0.09	
1 week	0.20	0.06	−0.91	0.15	−0.39	0.28	−0.50	0.36	−0.09	0.37	
1 month	1.54	0.09	0.15	0.05	1.32	0.32	1.22	1.63	0.59	0.16	
6 months	−1.06	0.06	2.16	0.16	3.60	0.57	3.76	0.38	1.72	1.37	
Mean	−0.13	1.12	0.03	1.35	1.29	1.53	0.82	2.11	0.37	1.16	

For the a* values of shade C1 (Table 3) at baseline, all values were again in the green zone at baseline except PFM. All ceramics progressively moved towards red zone till 6 months period with the greatest increase seen in ZrM. PFM shifted from red to green zone after 1 week but again shifted to red zone progressively till 6 months. Emax also shifted towards red zone till 1month but reverted back to green zone again after 6 months of exposure.

For the b* values of shade B1 (Table 4), all values were in the yellow spectrum at baseline and progressively increased in the positive values indicating an increased shift towards the yellow zone with time. The highest increase in b* value towards yellow zone was seen in PFM, (mean = 17.48).

Table 4 Standard deviation and mean of b* values of five test materials (shade B1 & C1).

Shade	Time interval	Emax	Lmax	PFM	ZrM	ZrL	
		Mean	S.D	Mean	S.D	Mean	S.D	Mean	S.D	Mean	S.D	
B1	Baseline	9.35	0.24	7.39	0.44	11.81	0.41	10.34	0.49	10.00	0.80	
1 week	14.35	0.26	9.49	0.52	17.62	1.77	12.82	0.46	11.33	0.39	
1 month	9.72	0.36	12.64	0.32	14.70	0.83	15.55	0.78	14.16	0.70	
6 months	25.12	1.62	29.50	2.57	25.80	2.99	26.82	1.10	22.44	1.38	
Mean	14.63	6.47	14.76	8.88	17.48	5.55	16.38	6.40	14.48	4.95	
C1	Baseline	5.03	0.15	6.62	0.75	14.60	0.91	10.77	0.53	8.82	0.46	
1 week	12.95	0.84	7.41	0.60	12.37	0.63	10.85	0.50	10.76	0.66	
1 month	19.78	0.37	14.04	1.77	18.80	0.51	19.83	5.16	15.43	0.98	
6 months	9.57	0.42	21.65	0.57	27.59	1.07	27.11	1.17	16.98	5.85	
Mean	11.83	5.43	12.42	6.19	18.34	5.92	17.14	7.37	13.00	4.44	

For the b* values of shade C1 (Table 4), all the specimens were in the yellow zone at base line. The b* value of Emax increased at 1 week and 1 month of exposure then dropped down to 9.57 at 6 months from a value of 19.78 at 1 month. All other 4 specimens demonstrated a progressive increase in b* value with time.

The mean of difference in shade ΔE for shade B1 is given in (Table 5). All specimens underwent an increase in ΔE well above the clinically discernible value of 3.3. The highest change in shade was exhibited by layered ceramic, Lmax, whereas the least change in shade was exhibited by Emax and then PFM at values of ΔE = 10.11 and 11.2 respectively. Shade B1’s color alterations, as shown in Table 5 and Fig. 5, were statistically significant (P < 0.05) across all five test materials at various smoke exposure times.

Table 5 Mean of difference in shade (ΔE) for five test specimens (shade B1).

Time interval	Emax	Lmax	PFM	ZrM	ZrL	Anova p value*	
	Mean	S.D	Mean	S.D	Mean	S.D	Mean	S.D	Mean	S.D		
Baseline	0.00	0.00	0.00	0.00	0.00	0.00	0.00	0.00	0.00	0.00	0.000	
1 week	2.65	0.44	1.70	1.79	3.33	0.76	2.85	0.40	1.93	0.84	0.000	
1 month	7.30	0.86	2.65	0.21	4.40	0.82	3.09	0.05	5.25	0.32	0.000	
6 months	10.11	0.33	17.02	2.66	11.21	1.08	12.10	1.62	15.66	1.20	0.000	
Anova p value	0.000	0.000	0.000	0.000	0.000		
Notes.

* ANOVA values for comparison between each group at different intervals.

Figure 5 Mean of difference in shade (ΔE) for five test specimens (Shade B1).

The mean of difference in shade ΔE for shade C1 is given in (Table 6). Again, all specimens demonstrated a change in shade value ΔE above the clinically discernible value of 3.3. The highest change in shade at 6 months was demonstrated by monolithic Zirconia (11.47) whereas lowest values of ΔE were exhibited by Emax (7.52). The color variations for shade C1 as presented in Table 6 and Fig. 6 were statistically significant (P < 0.05) within each of the five test materials at different time intervals of smoke exposure.

Discussion

Newer ceramic materials have shown better aesthetic qualities than other dental restorative materials (Makkeyah et al., 2024). Two of the most critical properties of these dental materials are color stability and reduced stainability (De Matos et al., 2022; Makkeyah et al., 2024; Alandia-Roman et al., 2013; Kim & Kim, 2016). The color stability refers to the change in color of a dental material over time in saliva, whereas stainability is the color changes that occur due to staining by either extrinsic or intrinsic colorants (Alandia-Roman et al., 2013). This research evaluated the effect of conventional smoking on color stability of the available dental ceramics. A unique mode of exposure of disks to cigarette smoke was setup for this study. A specialized vacuum chamber was fabricated to simulate the lung directing smoke at the materials at a distance of 10 cm. A significant color change was observed in dental ceramic materials in this study.

It has been documented that cigarettes smoke can cause staining of natural dental hard tissues as well as restorative dental materials  (Alrabeah et al., 2023a; Blatz et al., 2019; De Matos et al., 2022; Warreth & Elkareimi, 2020; Lamont, Koo & Hajishengallis, 2018; Alrabeah et al., 2023b; Tanthanuch et al., 2022; World Health Organization, 2023; Karanjkar et al., 2023; Dalrymple et al., 2021; Thompson et al., 2016; Makkeyah et al., 2024; Alandia-Roman et al., 2013; Kim & Kim, 2016). One of the reasons for this phenomenon may be due to the sticking of combustion by-products of traditional cigarettes on the surfaces of teeth and restorative materials, which causes color changes (World Health Organization, 2023; Thompson et al., 2016; Alandia-Roman et al., 2013). Multiple factors like smoking and lack of oral hygiene can accelerate the ageing and deterioration of dental materials (Grech & Antunes, 2019). A previous study by Thompson et al. (2016) attributed the discoloration of ceramics in cigarette smokers to the large amounts of smoke residues on the surfaces of ceramic samples.

The color changes in the materials in this study were evaluated by a spectrophotometer (Labscan XE; Hunter Lab, Reston, VA, USA). Spectrophotometers are considered as the most accurate instruments for color matching and shade detection in dentistry, particularly when used with CIELAB color coordinate system (Alrabeah et al., 2023b; Tanthanuch et al., 2022; Alandia-Roman et al., 2013; Philippi et al., 2023). According to evidence-based research, when comparing observations by the human eye, or conventional techniques, spectrophotometers exhibit a 33% increase in accuracy and a more objective color match in 93.3% of cases (Philippi et al., 2023). The measurements of a spectrophotometer are not influenced by ambient light because it measures the amount of light directly reflected from the samples and collected throughout a complete spectrum of wavelengths  (Philippi et al., 2023). The common consensus on its accuracy for color research in dentistry is a measure of the validity of this methodology for this research.

In color research, perceptibility (PT) and acceptability (AT) thresholds are used for interpretation of color changes (Alrabeah et al., 2023a; Blatz et al., 2019; Alandia-Roman et al., 2013). This may aid in selection of esthetic dental materials via interpretation of visual and instrumental findings. It also may indicate the level of clinical performance of esthetic materials, under different intra-oral conditions. In dentistry, color change PT and AT have been found to be ΔE = 1.2 and ΔE = 2.7, respectively (Alrabeah et al., 2023a; Blatz et al., 2019; Thompson et al., 2016; Alandia-Roman et al., 2013; Philippi et al., 2023; Aljamhan et al., 2022).

ΔE is the total color difference as per the CIELAB system of color evaluation. It is considered a gold standard of measurements used in color research for assessment of color changes (Blatz et al., 2019; Alandia-Roman et al., 2013). ΔE < 1 cannot be seen with the unaided eye, ΔE < 3.3 can only be seen by an experienced dentist, and ΔE > 3.3 may be seen by laymen as well and is clinically undesirable (Makkeyah et al., 2024; Aljamhan et al., 2022; Alnasser et al., 2021).

Table 6 Mean of difference in shade (ΔE) for five test specimens (shade C1).

Time interval	Emax	Lmax	PFM	ZrM	ZrL	Anova p value *	
	Mean	S.D	Mean	S.D	Mean	S.D	Mean	S.D	Mean	S.D		
Baseline	0.00	0.00	0.00	0.00	0.00	0.00	0.00	0.00	0.00	0.00	0.000	
1 week	2.90	0.33	2.28	0.47	2.26	0.47	2.77	1.16	3.18	2.85	0.310	
1 month	6.09	0.37	3.41	0.62	3.54	0.33	5.17	3.62	5.45	1.13	0.000	
6 months	7.52	0.40	8.72	0.57	8.76	0.50	11.47	0.76	8.44	6.11	0.004	
Anova p value	0.000	0.000	0.000	0.000	0.000		
Notes.

* ANOVA values for comparison between each group at different intervals.

Figure 6 Mean of difference in shade (ΔE) for five test specimens (Shade C1).

In this study, five materials were evaluated, pressable ceramics (Emax), layered ceramics (Lmax), porcelain fused to metal (PFM), monolithic zirconia (ZrM) and layered zirconia (ZrL). Two shades were used for assessment of each material, B1 and C1. A change in shade was observed in all five materials, accompanied by lowering of value. According to a meta-analysis done in 2022 (Schelkopf et al., 2022) smoking of traditional cigarettes results in highest change in shade of dental ceramics as compared to e-cigarettes or heated tobacco products (HTP). This change in shade in dental ceramics due to traditional cigarette smoke has been observed to be less when compared to other dental materials exposed to the smoke, such as dental composites and acrylic (Zanetti et al., 2019; Schelkopf et al., 2022; Elebiary & Ezzelarab, 2023). Dental composites were shown to have the highest staining followed by acrylic (Zanetti et al., 2019; Paolone et al., 2022). The highest staining has been shown to be within enamel and dentine according to multiple studies (Zanetti et al., 2019; Schelkopf et al., 2022; Alandia-Roman et al., 2013). This may indicate that the staining of different dental materials due to cigarette smoke is dependent on the structure of the material (Elebiary & Ezzelarab, 2023). This finding has been confirmed by multiple studies (Alrabeah et al., 2023a; Zanetti et al., 2019; Schelkopf et al., 2022; Thompson et al., 2016; Alandia-Roman et al., 2013; Paolone et al., 2023; Sayed et al., 2024). The structure of glass ceramics such as pressable and layered ceramics (Emax and Lmax) consists of tightly packed crystals surrounded by a glass matrix (Alandia-Roman et al., 2013). This leads to superior aesthetics as compared to zirconia in which the glass matrix is missing, with tightly packed crystals constituting the main structure (De Matos et al., 2022; Alandia-Roman et al., 2013). This leads to a decreased reflective index of zirconia resulting in inferior or opaque esthetic when compared to glass ceramics (De Matos et al., 2022; Zanetti et al., 2019; Alandia-Roman et al., 2013).

For shade C1, all five materials had undergone a shade change with a value of ΔE above 3.3, meaning that this staining could be observed clinically and was undesirable. Out of these materials, lithium di silicate (Emax) and PFM showed the lowest change in shade. For shade B1, all five materials exhibited a value of ΔE above 3.3, with the lowest change occurring in lithium di silicate (Emax) and PFM and highest color change occurring in Lmax and ZrL. PFM discs had a comparatively higher color stability than other ceramics, which can be attributed to the glazing effect as well as the presence of metal, opaquer, and body/dentine porcelain. According to literature, restorations show considerably less color variations after glazing compared to mechanical polishing (Ghazaee et al., 2024). The glazed surface outperforms the mechanical polishing method in terms of color stability. Particle interaction with the surface can be influenced by the glaze’s composition and surface texture. Particle adhesion may be increased by a rougher or more porous coating, but it may be decreased by a smoother glaze (Ghazaee et al., 2024; Özdemir & Duymus, 2019). This finding is in agreement with another study, in which the effect of e-cigarettes was observed on the color stability of different ceramics (Zanetti et al., 2019). The lowest change in shade was exhibited by porcelain fused to metal, which is similar to the present study (Zanetti et al., 2019; Alandia-Roman et al., 2013). In multiple studies, lithium disilicate (Emax) has been shown to be more color stable to extrinsic stains as compared to other ceramics (Zanetti et al., 2019; Ibrahim & Hassan, 2021; Al Moaleem et al., 2022; Schelkopf et al., 2022; Thompson et al., 2016; Makkeyah et al., 2024; Alandia-Roman et al., 2013; Alnasser et al., 2021). This may be due to the inherent stability in its chemical structure as compared to other ceramics, particularly layered ceramics like Lmax in our study  (Thompson et al., 2016; Alandia-Roman et al., 2013; Pires-de et al., 2009; Stamenković et al., 2021). Both Emax and Lmax materials’ surfaces differ due to the difference in their compositions and fabrication techniques (https://www.ivoclar.com/en_li/products/metal-free-ceramics/ips-e.max-press-lab), with Emax pressed (IPS Emax) and Lmax pressed (IPS Emax) followed by layering (IPS InLine Dentin), which may have resulted in potential color variations.

In shade C1, the highest change in shade as indicated by the ΔE values of 4.85 and 4.27, were found in monolithic and layered zirconia respectively. This can be explained by some of the structural properties and composition of zirconia. Zirconia has a structural lattice that may exist as monoclinic crystals or tetragonal/cubic lattice (Al Moaleem et al., 2022). This structure may be prone to low temperature degradation under different clinical conditions, particularly in the presence of saliva. Saliva’s buffering capacity, pH regulation, and re-mineralization impact the durability and stability of materials. The salivary esterase and enzymes can break down the composite resin matrix and causes corrosion of amalgam (Alakkad et al., 2023). According to Palacios et al. (2021), zirconia has high sensitivity to saliva which produces its microstructural destabilization, going from tetragonal to monoclinic phase that is much more fragile. Despite compositional and microstructural strength of zirconia, it exhibits small gaps between grains and the matrix which results in the material degrading most intensely after immersion in saliva (Palacios et al., 2021). This degradation can lead to disintegration of the crystal lattice, further leading to micro cracks and surface roughness which reduces the translucency of an already opaque material and can also encourages the deposition of staining particles such as those found in cigarette smoke  (De Matos et al., 2022; Al Moaleem et al., 2022; Schelkopf et al., 2022; Thompson et al., 2016; Makkeyah et al., 2024; Alandia-Roman et al., 2013; Kim & Kim, 2016; Sayed et al., 2024). Based on our findings, the clinician should counsel and educate the patients desiring long term esthetic restorations against detrimental effects of smoking on the restorative materials.

Dental ceramic materials are often subjected to smoke exposure, which is challenging to simulate. Most studies use custom-made devices, such as the unique one used in this study, which consist of an isolated exposure chamber, smoke-producing device, and suction system (Paolone et al., 2022). Some are essential, like standard test tubes connected to vacuum cleaners, while others use glass chambers. The Vitrocell System, an advanced exposure system, allows samples lodged in metal inserts to be exposed uniformly, making it more reliable for in vitro smoke staining of specimens (Paolone et al., 2023). The limitation of this study was the in vitro design of the study due to which the exact intra-oral conditions such as moisture, saliva, temperature and various other causes of staining such as food or acid reflux could not be simulated. These intra-oral conditions could have their own effects on the color stability of dental materials. Another limitation of this study was that the disks were not exposed to any cleaning procedure like tooth brushing or use of other cleaning aids by the individuals or physical removal of debris to some extent by lips, tongue and cheeks. Cleaning could help in removing the stains to varying degree and prolong the esthetics of dental materials.

For future studies, the use of cleaning aids on color stability of ceramics or performing the research in vivo is recommended. Effects of different cigarette brands having different concentration of nicotine and eating habits should also be considered.

Conclusion

This in vitro study demonstrated that despite recent advances in dental ceramics, shade of all types of ceramic restorations can be affected by smoking cigarettes. Within the constraints of this study, it is advised that Emax (lithium disilicate) rather than zirconia be used when restoring aesthetics with anterior restorations for a habitual smoker.

Supplemental Information

Data S1 Data

Additional Information and Declarations

Competing Interests

Author Contributions

Ethics

Data Availability

The authors declare there are no competing interests.

Abdulaziz Alqahtani conceived and designed the experiments, prepared figures and/or tables, and approved the final draft.

Abdulaziz A. AlHelal performed the experiments, authored or reviewed drafts of the article, and approved the final draft.

Ragad Albani conceived and designed the experiments, performed the experiments, prepared figures and/or tables, and approved the final draft.

Mohsin Ali analyzed the data, authored or reviewed drafts of the article, and approved the final draft.

Omar Ali Omar Badghshar performed the experiments, prepared figures and/or tables, and approved the final draft.

Aleshba Saba Khan analyzed the data, prepared figures and/or tables, authored or reviewed drafts of the article, and approved the final draft.

Syed Rashid Habib conceived and designed the experiments, analyzed the data, prepared figures and/or tables, and approved the final draft.

The following information was supplied relating to ethical approvals (i.e., approving body and any reference numbers):

The research was carried out at King Saud University’s College of Dentistry’s Department of Prosthodontics and CDRC in Riyadh, Saudi Arabia. Approval from ethical board of the University was obtained before starting the research work (Registration No PR-0128).

The following information was supplied regarding data availability:

The raw data is available in the Supplemental File.

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
