# Peer review of "Effect of cigarette smoking on the optical properties of contemporary dental ceramics: an in-vitro analysis"

_PeerJ, doi:10.7717/peerj.18564_

## Round 0.1 · original submission · Major Revisions

Please address all concerns of the reviewers and amend manuscript accordingly.

Reviewer 1 ·

Basic reporting

The. paper “Effect of Cigarette Smoking on the Optical Properties 1 of Contemporary Dental Ceramics: An In-Vitro Analysis” is a spectrophotometric analysis of the effect of cigarette smoking on the optical properties of ceramics using CIELAB coordinates. The aim is to help clinicians in the selection of esthetic materials that are resistant to staining for patients with the history of heavy cigarette smoking. 

The article covers a very interesting topic. Nevertheless, with some improvements I would consider the publication of this paper.

Was the normality of the data analyzed? What if the normality failed to be confirmed and a parametric test was used?

Why a simple ANOVA test was used and not a ANOVA for repeated measures was used?



Absence of a control group that is not exposed to cigarette smoke would make it harder to differentiate the effects of smoking from other factors. Please argument.

In the discussion the authors could mention the limits and the difficulty of a “smoke exposure” experimental design. The authors could reference to a couple of papers on smoke and color stability: PMID: 36602255  and 36070929 where several different procedures are described.

Experimental design

How were the specimens kept when not exposed to smoking? please specify

When spectrophotometric analysis was performed, the specimens were rinsed under water? Please specify.

Why the authors selected the CIELAB formula and not the CIEDE2000 (whch is more specific for dental color measurements) one?

Validity of the findings

Please add bar chart to understand visually the color change of different times and different materials

Reviewer 2 ·

Basic reporting

1- Some English revisions regarding the use of American and British English (the words “aesthetic” and “esthetic” have been used interchangeably several times. Kindly revise.
2- Adding graphs can be more visually descriptive than tables.

Experimental design

3- In the introduction, the following phrase stated, "Dental porcelains have been the material of choice for tooth colored indirect restorations for both posterior and anterior teeth”. I believe that you mean “dental ceramics” as “dental porcelains” are not recommended for posterior teeth.
4- In the introduction, it was mentioned that “The surface hardness determines the amount of time taken by tobacco to stain the specific restoration surface”, kindly explain how. It is confusing as the references mentioned for this statement did not investigate surface hardness and its effect on material stainability.
5- In tables 5 and 6, kindly remove the mean of color change at different time periods as it is confusing and has no indication.
6- What is the significance of investigating 2 different shade sof the materials

Validity of the findings

7- For B1 shade, explain why Lmax showed different results than Emax as the Lmax shows the highest change while the Emax showed the lowest.
8- In the results section, you mentioned that “Color variation was significant for all the materials”, kindly clarify if there are any homogenous groups that behave similarly or that the five groups were significant.

Annotated reviews are not available for download in order to protect the identity of reviewers who chose to remain anonymous.

Reviewer 3 ·

Basic reporting

Thank you for submitting such an interesting work.
The reviewer would like the author to review the following:
1- In materials and method:
- CDRC in Riyadh? WHAT DOES "CDRC" STAND FOR? Please mention.

2- Materials used:
- Why just 15 specimens? Has the author performed a sample size calculation? please elaborate and include it in the text.
- Please change the number 15 to "Fifteen".

3- Statistical test:
Please insert the full name before the abbreviation 'SPSS'.

4- Discussion:
- In 2nd Paragraph, please change by products to 'by-products'.
- in 4th Paragraph, please change ''help'' to ''aid"
- In paragraph #7, the author mentioned that the shade in the PFM group was more stable than the other groups, which was attributed to metal restoration on one side and porcelain on the other. Is it because of PFM glazing? In clinical situations and regardless of the type of ceramic restoration, one side of the restoration is subjected to smoke vapor (outer/glazed surface), while the other side is cemented and not subjected to smoke vapor. Why both surfaces of the restorations were subjected to smoke vapor? The reviewer thinks the current justification is insufficient and the discussion needs more elaboration and clarification for the obtained results. In addition, were these restoration discs subjected to a glazing step during fabrication and before testing? Please mention this in the methodology section as well.
- Regarding monolithic and layered zirconia, the author mentioned that the structure of the material is prone to low-temperature degradation under different clinical conditions, particularly in the presence of saliva. What is the rationale in this context?. Is it because of the saliva acidity? or just because of thermo-physical degradation of the restoration? It's important to explain the interaction between these factors and their relative contributions to the degradation process. More elaboration is required for this part of the discussion.

5- Conclusion:
Please revise the conclusion section. Instead of reiterating the results, the conclusion section should emphasize the study's broader implications and offer concrete recommendations.

Experimental design

Thank you for submitting such an interesting work.
The reviewer would like the author to review the following:
1- In materials and method:
- CDRC in Riyadh? WHAT DOES "CDRC" STAND FOR? Please mention.

2- Materials used:
- Why just 15 specimens? Has the author performed a sample size calculation? please elaborate and include it in the text.
- Please change the number 15 to "Fifteen".

3- Statistical test:
Please insert the full name before the abbreviation 'SPSS'.

4- Discussion:
- In 2nd Paragraph, please change by products to 'by-products'.
- in 4th Paragraph, please change ''help'' to ''aid"
- In paragraph #7, the author mentioned that the shade in the PFM group was more stable than the other groups, which was attributed to metal restoration on one side and porcelain on the other. Is it because of PFM glazing? In clinical situations and regardless of the type of ceramic restoration, one side of the restoration is subjected to smoke vapor (outer/glazed surface), while the other side is cemented and not subjected to smoke vapor. Why both surfaces of the restorations were subjected to smoke vapor? The reviewer thinks the current justification is insufficient and the discussion needs more elaboration and clarification for the obtained results. In addition, were these restoration discs subjected to a glazing step during fabrication and before testing? Please mention this in the methodology section as well.
- Regarding monolithic and layered zirconia, the author mentioned that the structure of the material is prone to low-temperature degradation under different clinical conditions, particularly in the presence of saliva. What is the rationale in this context?. Is it because of the saliva acidity? or just because of thermo-physical degradation of the restoration? It's important to explain the interaction between these factors and their relative contributions to the degradation process. More elaboration is required for this part of the discussion.

5- Conclusion:
Please revise the conclusion section. Instead of reiterating the results, the conclusion section should emphasize the study's broader implications and offer concrete recommendations.

Validity of the findings

Thank you for submitting such an interesting work.
The reviewer would like the author to review the following:
1- In materials and method:
- CDRC in Riyadh? WHAT DOES "CDRC" STAND FOR? Please mention.

2- Materials used:
- Why just 15 specimens? Has the author performed a sample size calculation? please elaborate and include it in the text.
- Please change the number 15 to "Fifteen".

3- Statistical test:
Please insert the full name before the abbreviation 'SPSS'.

4- Discussion:
- In 2nd Paragraph, please change by products to 'by-products'.
- in 4th Paragraph, please change ''help'' to ''aid"
- In paragraph #7, the author mentioned that the shade in the PFM group was more stable than the other groups, which was attributed to metal restoration on one side and porcelain on the other. Is it because of PFM glazing? In clinical situations and regardless of the type of ceramic restoration, one side of the restoration is subjected to smoke vapor (outer/glazed surface), while the other side is cemented and not subjected to smoke vapor. Why both surfaces of the restorations were subjected to smoke vapor? The reviewer thinks the current justification is insufficient and the discussion needs more elaboration and clarification for the obtained results. In addition, were these restoration discs subjected to a glazing step during fabrication and before testing? Please mention this in the methodology section as well.
- Regarding monolithic and layered zirconia, the author mentioned that the structure of the material is prone to low-temperature degradation under different clinical conditions, particularly in the presence of saliva. What is the rationale in this context?. Is it because of the saliva acidity? or just because of thermo-physical degradation of the restoration? It's important to explain the interaction between these factors and their relative contributions to the degradation process. More elaboration is required for this part of the discussion.

5- Conclusion:
Please revise the conclusion section. Instead of reiterating the results, the conclusion section should emphasize the study's broader implications and offer concrete recommendations.

Additional comments

Thank you for submitting such an interesting work.
The reviewer would like the author to review the following:
1- In materials and method:
- CDRC in Riyadh? WHAT DOES "CDRC" STAND FOR? Please mention.

2- Materials used:
- Why just 15 specimens? Has the author performed a sample size calculation? please elaborate and include it in the text.
- Please change the number 15 to "Fifteen".

3- Statistical test:
Please insert the full name before the abbreviation 'SPSS'.

4- Discussion:
- In 2nd Paragraph, please change by products to 'by-products'.
- in 4th Paragraph, please change ''help'' to ''aid"
- In paragraph #7, the author mentioned that the shade in the PFM group was more stable than the other groups, which was attributed to metal restoration on one side and porcelain on the other. Is it because of PFM glazing? In clinical situations and regardless of the type of ceramic restoration, one side of the restoration is subjected to smoke vapor (outer/glazed surface), while the other side is cemented and not subjected to smoke vapor. Why both surfaces of the restorations were subjected to smoke vapor? The reviewer thinks the current justification is insufficient and the discussion needs more elaboration and clarification for the obtained results. In addition, were these restoration discs subjected to a glazing step during fabrication and before testing? Please mention this in the methodology section as well.
- Regarding monolithic and layered zirconia, the author mentioned that the structure of the material is prone to low-temperature degradation under different clinical conditions, particularly in the presence of saliva. What is the rationale in this context?. Is it because of the saliva acidity? or just because of thermo-physical degradation of the restoration? It's important to explain the interaction between these factors and their relative contributions to the degradation process. More elaboration is required for this part of the discussion.

5- Conclusion:
Please revise the conclusion section. Instead of reiterating the results, the conclusion section should emphasize the study's broader implications and offer concrete recommendations.

---

## Round 0.2 · accepted · Accept

All issues pointed out by the reviewers were addressed and the revised manuscript is acceptable now.

Reviewer 1 ·

Basic reporting

The authors have provided all the requested improvements.

Experimental design

Ok.

Validity of the findings

Ok.

Additional comments

Ok.